# Rupture of the Pleomorphic Adenoma of the Parotid Gland: What to Know before, during and after Surgery

**DOI:** 10.3390/jcm10225368

**Published:** 2021-11-18

**Authors:** Michele Grasso, Massimo Fusconi, Fabrizio Cialente, Giulia de Soccio, Massimo Ralli, Antonio Minni, Griselda Agolli, Marco de Vincentiis, Marc Remacle, Paolo Petrone, Domenico Di Maria, Vito D’Andrea, Antonio Greco

**Affiliations:** 1Department of Sense Organs, Sapienza University of Rome, 00161 Rome, Italy; Massimo.fusconi@uniroma1.it (M.F.); fabrizio.cialente@uniroma1.it (F.C.); giulia.desoccio@uniroma1.it (G.d.S.); massimo.ralli@uniroma1.it (M.R.); antonio.minni@uniroma1.it (A.M.); griselkena@yahoo.it (G.A.); antonio.greco@uniroma1.it (A.G.); 2Department of Oral and Maxillofacial Sciences, Sapienza University of Rome, 00161 Rome, Italy; marco.devincentiis@uniroma1.it; 3Department of Otorhinolaryngology Head and Neck Surgery, CHL-Eich, Rue d’Eich 78, 1111 Luxembourg, Luxembourg; marc.remacle01@gmail.com; 4Department of Otolaryngology, Head and Neck Surgery, “Di Venere” Hospital, 70121 Bari, Italy; info@paolopetrone.it; 5Department of Otolaryngology, Head and Neck Surgery, “San Pio” Hospital, 82100 Benevento, Italy; domenico.dimaria@ao-rummo.it; 6Department of Surgical Sciences, “Sapienza” University of Rome, Viale Regina Elena 324, 00161 Rome, Italy; vito.dandrea@uniroma1.it

**Keywords:** superficial parotidectomy, pleomorphic adenoma, parotid capsule, recurrence of pleomorphic adenoma, intraoperative spillage

## Abstract

Background: We assessed the cases of intraoperative spillage of primary pleomorphic adenomas (PPAs) of the parotid gland in the literature, comparing them with our own cases. We aim to explain how the surgeon should manage a spillage during surgery (i.e., how to avoid spreading the contents that are coming out of the tumor). We also aim to investigate whether or not spillage is linked to a higher rate of PPA recurrence. Methods: We collected surgical and pathological reports, taking data on capsular ruptures and the spillage of tumors. Results: Intraoperative tumor spillage and tumor rupture occurred in 34/202 cases. There were three recurrences after a mean of 3.7 years (mean follow-up duration: 10.3 years). One recurrence happened to a patient who had an intraoperative tumor spillage, and two more recurrences happened to patients who did not have spillage. Conclusion: We believe that the real number of the events of spillage is underestimated and underreported by surgeons. Capsular rupture must always be avoided, and secure resection margins must always be pursued, independent of the type of parotidectomy being performed. Features that increase the risk of recurrence are an intraoperative rupture and the presence of satellite nodules (as recorded in the pathologist’s report). In these cases, patients need a longer follow-up period.

## 1. Introduction

Pleomorphic parotid adenoma (PPA) is the most common parotid tumor (accounting for 60% to 70% of all parotid tumors) and it can occur at all ages (the highest incidence rate occurs between the 4th and 6th decades of life), although the incidence rate of cystadenolymphomas has increased over time. PPAs display cellular pleomorphism and are made of a component of epithelial and connective tissue that is inserted into a stroma of mucoid, myxoid, chondroid, or osteoid origin (or any mix of these origins). The most common way to treat a PPA that is less than 4 cm in diameter and located in the lateral lobe is superficial parotidectomy (SP), as this method has the greatest chance of preserving the facial nerve. PPAs treated with SP have a recurrence rate of 1% to 4%, although the rate of extracapsular dissection has been increasing in recent years [1,2,3,4,5,6,7,8,9,10,11,12]. The recurrence rates described in the different reports published in the literature depend strongly on the duration of the follow-up period, as well as the form of follow-up examination (e.g., palpation, sonography, or MRI). Tumor recurrence is closely associated with incomplete surgical excision, accidental rupture of the pseudotumor capsule during surgery, incomplete tumor capsule removal, and multicentricity, all of which allow the spillage of tumor cells into the wound. The literature confirms that a high percentage of PPA cases, those of the stroma-rich neoplasms, reveal a focal absence of the capsule and the formation of satellite nodules [1]. Colella et al. found that the importance of accidental capsule ruptures could be overestimated [1]. The authors suggested that the margins are only microscopically associated with higher recurrence, while intraoperative neoplasm leakage is not at all associated with higher recurrence. The real problem in parotid surgery is posed by the facial nerve. Due to its complex branching form and its connections within the parotid gland, a small tumor of the superficial lobe of the gland could meet a branch of the nerve. A complete resection would require the sacrifice of this branch, but this sacrifice, obviously, cannot be proposed.

## 2. Materials and Methods

Patient Population: Data were taken from 202 patients with a PPA of the superficial lobe of the parotid gland who were treated at the Department of Otolaryngology, Head and Neck Surgery, Sapienza University of Rome, between 2002 and 2009. A retrospective review was performed on these data. Patients who underwent superficial parotidectomy (SP) were enrolled in the study and their pathological reports confirmed the diagnosis of PPA (Table 1). The exclusion criteria for this study were having undergone total parotidectomy (either for PPA or another reason), undergoing immunosuppression treatment, or undergoing locoregional radiotherapy. At the time of the follow-up in 2021, all patients reported whether or not they had suspected a recurrence of the tumor and if they had been treated for any recurrence. The patients underwent regular follow-ups with clinical examination, ultrasound, and MRI if necessary.

Methods: The following patient characteristics: age, sex, and the duration of their follow-up period, were collected from 202 cases of PPAs that were treated with the surgical method of SP. We then collected data on the patients’ tumor location, tumor size, capsule rupture status, any instance of intraoperative tumor spillage, and the histopathological features of the tumor in order to relate these to the patients’ clinical outcomes (Table 2).

The tumors were categorized into 3 histological subtypes: the classic subtype with a stroma content between 30% and 50%; the stroma-rich subtype (myxoid) with a stroma content of 80%; and the cellular subtype with a stroma content between 20% and 30% (or less). Regarding the capsular features identified, ‘incomplete capsule’ describes a focal absence in one or more of the neoplasm capsule points, ‘pseudopodia’ describes neoplasm nodules that are ejected through the incomplete capsule but placed inside the neoplasm capsule, and ‘satellite nodules’ describe neoplasm nodules that are in proximity to the main neoplasm nodule although not connected to it. The tumor margin was defined positive when it was marked with ink, or the distance between the neoplasm and the ink was less than 1 mm. None of the patients received preoperative or postoperative radiotherapy. This study was approved by the Institutional Ethical Committee of Sapienza University of Rome.

## 3. Results

The mean age of the patients at the time of surgery was 46.2 years. The cohort comprised 49 males and 153 females. The mean diameter of the treated tumors was 1.9 cm. The mean duration of the follow-up period was 11.2 years (min 9.2–max 17.2 years). All of the patients underwent preoperative FNAC (fine needle aspiration cytology), which returned a reading that was positive for PPA in 197 cases and a reading that was suspicious of PPA in 5 cases. All of the tumors were superficial lobe PPAs. In 15 of the 202 cases, a positive margin on the ink margin was identified. In 24 of the 202 cases, the surgical sample was fragmented. Intraoperative tumor spillage and tumor rupture occurred in 34 out of the 202 cases. The group of patients with intraoperative tumor spillage comprised 5 men and 29 women; the PPAs of these patients had a mean diameter of 1.9 cm. We observed three relapses in the cohort of PPA patients who were treated with SP after a mean period of 3.7 years (mean FU 10.3 years). One of these recurrences was from among the group of patients who had an intraoperative tumor spillage occur and the other two were from among the group of patients who did not have an intraoperative tumor spillage. We did not find a statistically relevant difference in terms of the rate of relapse between cases with or without tumor rupture and spillage (1/34 vs. 2/168, respectively). The mean diameter of the relapsed tumors was 2.34 cm. Among these three patients who experienced a relapse, those with spillage had a mean tumor diameter of 1.6 cm, while those without spillage had a mean tumor diameter of 1.9 cm. In two out of the three relapsed cases, the tumor capsule was incomplete, and in one case, there was an intraoperative spillage of the tumor. Among the cases of PPA recurrence, the average duration of the SP surgery was 35 min, which is a lower than the average time for this type of surgery. The overall average duration of the SP surgery (including revisions) for the whole cohort was 71.3 min. We did not observe any patients with permanent facial nerve dysfunction. Frey syndrome occurred in nearly 30% of the patients.

## 4. Discussion

Given the multifocal nature of recurrent PPAs, radicality of the surgery is difficult to achieve (Table 3). The rate of recurrence after surgery on benign salivary gland PPAs ranges between 20% and 45% after simple ECD, and between 2% and 5% after lateral lobectomy of the parotid gland, but it is only 0% to 0.4% after TP surgery. The location of the neoplasm in the deep portion of the parotid gland, large neoplasm size (2 cm or more, advanced patient age, and the neoplasm’s contact with the facial nerve are other factors that are suspected to increase the risk of recurrence [5]. Numerous authors have stated that microscopically positive margins are linked to a higher rate of recurrence, while intraoperative spillage is not [1,10,11]. Maynard JD [3] described one case of the recurrence of a 0.7 cm PPA under the surgical scar, at a mandibular angle, in a 29 year-old female, 11 years after surgery for a 4 cm-deep lobe PPA. In this case, the pathologist reported capsular rupture, after which, the surgical wound was irrigated with sterile water. Armistead [4] reported a population of 76 patients who underwent extracapsular dissection followed by RT. A total of 16 out of the 76 patients had a capsular rupture. The lateral lobectomy surgery (with dissection of the facial nerve branches) is considered by many authors [5] as the best type of dissection in most parotid-related cases. Henriksson et al. assessed cases of capsular rupture and showed no increased rate of recurrence in patients with macroscopic spillage during surgery compared to those without. PPAs, unlike other salivary gland neoplasms, do not have a true fibrous capsule. This allows the fingerlike branches of the tumor tissue to extend outside of the main tumor lump. According to Henriksson G et al. [5], the presence of such branching is significantly linked to a higher recurrence risk. This finding justifies the use of a more complete type of surgery to manage PPAs, i.e., a superficial lobectomy and removal of a part of normal parotid tissue.

SP should be contemplated for use as a partial capsular dissection and partial enucleation method. The real margins of the neoplasm during SP are the facial nerve or fascia that are superficial to the parotid gland [6]. In a study published by Witt et al. [6], the neoplasms studied were mobile PPAs of the superficial lobe, less than four cm in diameter. In that study, pathologists recorded a focal capsular exposure in 95% of PPA treated with TP, 100% in those treated with PSP, and 100% in those treated with ECD. Tumor spillage and tumor ruptures happened in 10% of the TP surgeries, 5% of the PSP surgeries, and 5% of the ECD surgeries. The authors did not observe recurrences in the patients who had undergone TP (with a mean FU of eight years), PSP (with a mean FU of eight years), or ECD (with a mean FU of nine years). Capsular ruptures were more severe when enucleation was performed, although the rate of capsular ruptures was equivalent among the procedures that were studied. The authors suggested that the presence of pseudopodia outside of the capsule is a risk factor for recurrence. Natvig K et al. [7] studied 238 patients who had been treated for PPA. Within this cohort, 40 patients underwent a subtotal parotidectomy, and 193 patients underwent a lateral lobectomy. Six patients had recurrences between 7 and 18 years post-operatively (mean: 11.8 years). Rupture of the capsule with a macroscopic spillage of tumor cells occurred in 26 patients, and 2 of them (8%) had recurrences. A surgical dissection close to the capsule was performed in 87 cases; in these cases, there was one recurrence. In 121 of the patients, the dissection was carried out without visualizing the tumor capsule, with three of these patients developing recurrence (2.5%). A microscopic-level analysis of the six patients who experienced recurrences showed one rupture of a capsule with positive margins, one tumor cell growing through a capsule with positive surgical margins, one close but negative surgical margin, two obviously negative surgical margins, and one surgical margin that was not described. They observed three recurrences in tumors with epithelial cells dominating, and three in tumors with epithelial and mesenchymal cells almost equally distributed. The average observation time in the aforementioned study was 18 years, which is longer than those utilized in other studies. Laccourreye H et al. [8] treated 229 cases of primary PPA using TP with facial nerve preservation. They observed one case of recurrence within their F-U for a minimum of 10 years. In the one case of recurrence, the authors speculated that the TP might have been incomplete due to the patient being 35 kg overweight. In this particular case, the local recurrence was a 2 by 1 cm unique mass located by the inferior border of the posterior belly of the digastric muscle. Inadvertent PPA spillage occurred in 22 patients. The authors concluded that tumor spillage should not be considered to be a factor that is related to tumor control in patients who are treated using TP. Certainly, the lateral and total parotidectomy procedures require a greater level of manipulation of the facial nerve and, therefore, have a higher incidence rate of causing its dysfunction. Total parotidectomy was associated with a significantly higher incidence rate of facial nerve dysfunction during the first postoperative period (60.5% at day 1 and 44.7% at month 1) than superficial parotidectomy (18.2% at day 1 and 10.9% at month 1) [12]. Gi Cheol Park et al. [9] observed that a perioperative rupture of the tumor was observed in three recurrent group subjects (30%), but only four of the non-recurrent group subjects (4.0%). Multivariate analyses showed that the risk of recurrence was more than 5-fold higher when satellite nodules were present and more than 14-fold higher when the tumor had ruptured. The authors concluded that both satellite nodules and tumor ruptures increased the risk of recurrence in patients with PPAs that were treated by SP. In the study of Gi Cheol Park et al. [9], authors found an incomplete capsule in 65/110 patients studied (59.1%). Regarding histological subtype, an incomplete capsule was observed in 2/7 cellular-variety neoplasms, 49/82 classic-variety neoplasms, and 14/21 myxoid-variety neoplasms. An incomplete capsule was detected in 19/25 neoplasms that were less than two cm in diameter, 39/70 neoplasms that were between two and four cm in diameter, and 7/15 neoplasms that were larger than four cm in diameter. The histological type and neoplasm dimensions of the PPA were not associated with an incomplete capsule. Pseudopodia were found in 53.6% of patients. Satellite nodules were found in 16 patients (14.5%). Univariate and multivariate analyses showed that satellite nodules were a significant risk factor for the recurrence of PPAs. Both analyses showed a correlation between neoplasmic rupture and PPA recurrence. Univariate analysis, but not multivariate analysis, showed an association between a positive resection margin and PPA recurrence. Focal capsular exposure was observed in 80% of PPAs in the prospective series, most often because of the close anatomic relationship between the tumor and the facial nerve’s branches. One or more capsular characteristics (i.e., incomplete capsule, capsular penetration, pseudopodia, and satellite nodules—all of which may compromise a complete tumor resection in several surgical techniques) were detected in 73% of the 218 PPAs studied. Larger tumors presented more frequently with satellite nodules than smaller tumors. Pseudopodia and satellite nodules appeared to be more common than presumed by other authors [13]. Intraoperative tumor spillage was not found to aggravate the prognosis, even in the case of the resection of a subsequent recurrence of pleomorphic adenoma (srPA) after surgery for the first recurrence [14]. McGurk et al. presented a series of 487 patients with “simple” parotid pleomorphic adenomas, i.e., PPAs that were discrete, mobile, and less than 4 cm in diameter (the same as all the cases presented in this study); 75% of these patients underwent an extracapsular dissection and 25% underwent a superficial parotidectomy. The median follow-up time was 12 years (range: 5–32 years). The rate of recurrence was 1.7% in patients treated with extracapsular dissection, and 1.8% in those treated with superficial parotidectomy [15]. Mantsopoulos et al. [16] presented a series of 637 cases of pleomorphic adenomas; 455 underwent an extracapsular dissection. Recurrences were observed in only four patients (0.9%). O’Brien et al. [17] presented 254 cases of PPA. Limited superficial parotidectomy was carried out in all previously untreated patients with tumors that were superficial to the plane of the facial nerve. There were only three cases of tumor recurrence (0.8%) in this series.

### 4.1. What to Know before Parotid Surgery for PPA

Imaging and cytology can help in the diagnosis of PPAs. Ultrasonography shows that PPAs are usually well defined, lobulated, hypoechoic lesions with posterior acoustic enhancement; a certain homogeneity is frequently reported. They can include focal calcification and are usually poorly vascularized, although increased vascularization is also recognized. PPAs are usually poorly vascularized by capsular and/or internal vessels [18]. Through the use of contrast-enhanced ultrasound (CEUS), a significantly stronger enhancement of perfusion (an increase in Doppler signal area) has been noted in PPAs. The most appropriate and cost-effective imaging modality is ultrasonography with a multiparametric approach (MPUS), which includes CEUS and elastography. When the assessment is still indefinite after the use of MPUS, the patient could be referred to MR imaging [18]. Ultrasound evaluation consists of B-mode, color-Doppler, and quasistatic ultrasound elastography (USE), conducted using the elasticity contrast index (ECI) elastography technique. The USE with ECI index measurements can generally discriminate preoperatively benign from malignant neoplasms, except for in the case of a PPA, which is stiff. One should be aware that PPAs are stiffer than other benign lesions [19]. FNAC may induce hemorrhage, infarction, and perversion of histological detail [2]. Neoplasms in their earlier stages of growth are more cellular because the epithelial component is more prevalent. Recent findings suggest that there is a higher rate of incomplete encapsulation in hypocellular neoplasms, in which the chondromyxoid stroma predominates. This softer, more friable hypocellular neoplasm could lead to rupture and spillage. In the same neoplasm, we can frequently observe areas of the capsule with thickness, few cells, or no capsule. Capsular penetration and perforation can happen where the neoplasm makes nodular protrusions into the complete capsule. Narrow-necked projections of neoplasm, particularly those that are deep in the parotid gland, could easily be interrupted during enucleation, leading to relapse. If imaging is suspect for a diagnosis of a PPA, it is worthwhile to consider leaving the healthy gland tissue that is surrounding the tumor in order to prevent damage to the capsule. This could be difficult if the tumor is very close to the facial nerve.

### 4.2. What to Do in Case of an Intraoperative Rupture of a PPA

It is imperative to stop the spread of the tumor with suction and, if possible, put some stitches in the healthy gland tissue surrounding the tumor capsule. Maynard JD et al. reported a series in which patients underwent postoperative RT (up to the early 1970s) or washing of the wound with sterile water. According to Webb AJ et al. [2], field irrigation is suggested to reduce the risk of tumor seeding. A very important aspect of PPA surgery is the actions required if there is an accidental rupture. Many articles include a variety of agents for irrigation of the wound, such as 0.15% cetrimide, normal saline, and sterile distilled water alternating with saline. Although the practice of irrigation of the wound is rare, there is an agreed upon general preference for liberal sterile water irrigation acting as a tumoricidal agent followed by normal saline (200–500 mL), repeated in the form of 20 mL syringe washings. Technical finesse in exactly identifying the plane of the facial nerve, together with a gentle, careful dissection, could avoid spillage and the separation of pseudopodia from the neoplasm, or a tumor rupture when the neoplasm is close to the nerve. It is hard to implant a PPA tumor. Irrigation after the excision is successful in taking away neoplasm remnants. The management of an intraoperative tumor rupture and the resultant leakage is a controversial topic (i.e., “to rinse or not to rinse”). For that reason, these statements should be clarified in big metanalyses or multicentric studies with a prolonged follow-up period (at least 10 years).

### 4.3. What to Do after Surgery for PPA

Clear surgical reporting, describing any macroscopic rupture of the capsule of a PPA resulting in spillage, is mandatory. The choice, and the report, of the remediation technique is important to reduce the spillage of the tumor. It is important to describe whether the PPA was adherent to the facial nerve or if there was healthy tissue between the facial nerve and capsule of the PPA. Patients should be carefully followed-up for many years, since in these cases, the rate of recurrence rises. The mean time to an incident of recurrence reported in the literature is 7 years. Late recurrences can happen after ten years.

## 5. Conclusions

The surgeon should always report a rupture of the capsule of a PPA and the pathologist must always describe this event in their report. The PPA must be removed, leaving a few millimeters of healthy glandular tissue intact. The rate of recurrence of PPAs rises when microscopic fingerlike formations of neoplasmic tissue run into the gland. This could explain why the use of more extensive types of surgery could reduce the rate of recurrence. If these neoplasmic extensions are identified by the pathologist, a long period of FU with the patient is necessary to check for the relatively small number of cases with local recurrence. When a PPA is handled with SP, it is mandatory to achieve a clear resection margin and prevent capsular rupture. When rupture happens, or the postoperative pathological report shows the presence of satellite nodules, a meticulous and extended FU period is needed, since these findings raise the rate of recurrence. The analysis of the presented series and the results of most of the previous studies of PPA treatment strengthen the hypothesis that a tumor rupture with spillage does not necessarily indicate an unfavorable prognosis. Further studies utilizing wider case histories are necessary to confirm this hypothesis.

## Figures and Tables

**Table 1 jcm-10-05368-t001:** Surgical techniques.

Name of Surgery	Description
*Total parotidectomy (TP)*	Removes all parotid tissue lateral and medial to the facial nerve.
*Superficial parotidectomy (SP)*	Removes the parotid tissue lateral to the facial nerve. SP dissects less than the full facial nerve; neoplasm is removed with two cm of normal parotid tissue.
*Extracapsular dissection (ECD)*	Dissection performed without previous identification of the facial nerve; a 2 to 3 mm edge of healthy gland tissue is taken away with the neoplasm.
*Partial superficial parotidectomy (PSP)*	This dissection includes less than the entire facial nerve and removes a large part (minimum margin of 2 cm, except when the tumor leans against the facial nerve) of the surrounding parotid tissue but does not eradicate healthy parotid tissue far from the neoplasm. Ideal candidates are patients with small, mobile, benign, or malignant, low-grade tumors (less than 4 cm in diameter), predominantly involving the superficial lobe.

**Table 2 jcm-10-05368-t002:** Patient population.

*Age, Years*		*Percentage %*
*Mean*	46.2	
*Range*	19–80	
*Sex*		
*Male*	49	*24*
*Female*	153	*76*
** *Tumor size, cm* **		
*≤2*	133	*66*
*2–4*	62	*31*
*≥4*	7	*3*
** *Pathological subtype* **		
*Cellular type*	32	*16*
*Classic type*	125	*62*
*Myxoid type*	45	*22*
** *Capsule* **		
*Complete*	114	*56*
*Incomplete*	48	*24*
*Pseudopodia*	40	*20*
** *Resection margin* **		
*Negative*	187	*93*
*Positive*	15	*7*
** *Intraoperative spillage of tumor* **		
*Absent*	168	*83*
*Present*	34	*17*
** *Follow-up, years* **		
*Median*	11.2	
*Range*	9.2–17.2	

**Table 3 jcm-10-05368-t003:** Literature reports.

Author	Total N° of Cases	Cases of Recurrences	Capsular Rupture	Notes
de Vincentiis M *(present study)*	202 PPA	3 pts	34 pts	
Maynard JD. [3]	114 PPA	0.7 cm nodule under the scar in 1 pt (29 year-old) after 11 years post a TP for a deep lobe 4 cm PA with capsular rupture. Wound irrigated with sterile water	8 pts out of 114	8 pts with capsular rupture underwent postoperative RT (up to early 1970s) or washing of the wound with sterile water
Armistead PR. [4]	76 PA treated with extracapsular dissection followed by RT	1 pts	16 pts	
Henriksson G et al. [5]	197 pts with PA	9 pts	28 pts	7 pts had recurrence without intraoperative capsule rupture, while 2 pts had recurrence with intraoperative capsule rupture
Robert L. Witt. [6]	60 PPA	There have been no recurrences in the patients with PPA treated by TP (mean FU, 8 years), in those treated with PSP (mean FU, 8 years), or in those treated with ECD (mean FU, 9 y)	Focal capsular exposure on histological examination occurred in 59 of 60 cases of PPA. Tumor spill and tumor rupture occurred in 10% of the cases of TP, 5% of the cases of PSP, and 5% of cases of ECD.	Focal capsular exposure was identified in numerous cases in which this finding was not mentioned in the pathology report.
Natvig K. [7]	238 PPA	6 pts from 7 to 18 years post-operatively (mean 11.8 years)	26 pts, 2 of them (8%) had recurrences	Surgical dissection close to the capsule was performed in 87 cases, with 1 recurrence. In 121 pts, the dissection was carried out without visualizing the tumor capsule, with 3 of them developing recurrence.
Laccourreye H. [8]	229 PPA treated with total parotidectomy with facial nerve preservation	1 case with 10 years FU. Parotidectomy might have been incomplete due to the patient being 35 kg overweight. Local recurrence was a 2 by 1 cm unique mass located by the inferior border of the posterior belly of the digastric muscle.	Inadvertent PA spillage occurred in 22 pts.	Tumor spillage should not be considered as a factor related to tumor control in pts treated with total conservative parotidectomy
Gi Cheol Park et al. [9]	110 pts with PPA treated with superficial parotidectomy	10 pts	65 pts	

## Data Availability

Data supporting reported results can be found in the databases of the authors.

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
