# Peer review of "Rupture of the Pleomorphic Adenoma of the Parotid Gland: What to Know before, during and after Surgery"

_jcm, 2021, doi:10.3390/jcm10225368_

Round 1

Reviewer 1 Report

Pleomorphic adenomas may recur after a long period of time. Long term is not defined, but relapse may occur after 20 years or more. I have experienceed many cases of pleomorphic adenoma at my institution. In my facility's experience, out of 304 cases of pleomorphic adenoma that were operated on, there were 13 cases of relapse, including 5 cases that recurred more than 20 years after the initial surgery. Thus, when discussing the relapse of pleomorphic adenoma, a long period of observation is necessary. Considering the need of a long observation, the observation period in this paper is short. Therefore, this paper should be written after another 10 years of observation.

Author Response

Dear Revisor,

we read with much attention your comments and thank for it. We believe that relapse of PPA may depend on surgical techniques (Total parotidectomy, Superficial parotidectomy, Extracapsular dissection, Partial superficial parotidectomy), on location of the tumor (superficial or deep lobe), on dimension of the PPA. Relapse of PPA is not only time dependent.

In our paper, we assessed only patients who underwent superficial parotidectomy, and the mean duration of follow-up was 11.2 years (min 9.2 - max 17.2 years). We believe that our follow up time is adequate. As we stated in the paper “The rate of tumor recurrence after surgery for benign salivary gland PPA ranges between 20% and 45% after simple ECD and between 2% and 5% after lateral lobectomy of the parotid gland, but it is only 0% to 0.4% when the more radical TP technique is used. Location of the tumor in the deep portion of the parotid gland, large tumor size, advanced patient age, and tumor contact with the facial nerve are other factors suspected to increase the risk of recurrence”.

In our paper, the mean diameter of the treated tumors was 1.9 cm. All tumors were superficial lobe PPAs.

Laccourreye H and other Authors reported a Follow Up period for a minimum of 10 years.

Reviewer 2 Report

There should be a citation of papers with recurrent PPA caused bei FNAC, because all pts underwent FNAC and it should be excluded as a cofactor for the examined outcomes.

238 … Cytology…(instead of citology)

Author Response

FNAC is a widely used method to extract material (cells) from neoplasms without disseminating the content. FNAC is also widely used for suspected malignant metastases to the neck nodes.

In our study we assessed patients who underwent superficial parotidectomy for PA of the superficial lobe. The route that the fine needle crosses to the pleomorphic adenoma includes a part of the gland that will be removed with surgery. Even if some pleomorphic adenoma cells spread into the healthy parenchyma of the gland, they would still be included in the part of the gland that is removed.

FNAC can’t lead to recurrent PPA. I have not found articles in the literature describing recurrent PPA caused by FNAC!

Reviewer 3 Report

A very important and interesting article on the risk of recurrence of PPA - a problem that is particularly difficult in patients who require reoperation. All the characteristics of PPA that increase the risk of local recurrence and the surgical aspects of rupture of the tumor capsule during surgery have been properly presented and discussed. The presented topic is always relevant.

Author Response

Dear Revisor,

thank you

Round 2

Reviewer 1 Report

I have already rejected this manuscript.

Author Response

Thank  you again for your comments.